# The Nickel: A History of African-Descended People in Houston's Fifth Ward

**Denise Frazier**

New Orleans Center for the Gulf South, Tulane University, New Orleans, LA 70118, USA; dfrazier@tulane.edu

**Abstract:** This paper will chronicle the unique stories that have come to exemplify the larger experience of Fifth Ward as a historically African American district in a rapidly changing city, Houston. Fifth Ward is a district submerged in the Southern memory of a sprawling port city. Its 19th century inception comprised of residents from Eastern Europe, Russia, and other religious groups who were fleeing persecution. Another way to describe Fifth Ward is much closer to the Fifth Ward that I knew as a child—an African American Fifth Ward and, more personally, my grandparents' neighborhood. The growing prosperity of an early 20th century oil-booming Houston had soon turned the neighborhood into an economic haven, attracting African Americans from rural Louisiana and east Texas. Within the past two decades, Latino communities have populated the area, transforming the previously majority African American ward. Through a qualitative familial research review of historic documents, this paper contains a cultural and economic analysis that will illustrate the unique legacies and challenges of its past and present residents. I will center my personal genealogical roots to connect with larger patterns of change over time for African Americans in this distinct cultural ward.

**Keywords:** Fifth Ward; migration; economics; culture; gulf south; memory

---

there is a phantom language in my mouth.
a tongue beneath my tongue.
will i ever
remember what
i sound
like.
will i ever come home
"african american i" from *salt* by nayyirah waheed
*
A good tree cannot bear bad fruit, and a bad tree cannot bear good fruit.
Matthew 7:18 (Waheed 2013)

## 1. Access

"I'm not sending you any instructions on how to make gumbo." I knew I should not have asked my aunt, Sandra Harris, to give me her seafood gumbo recipe via text message, or even over the phone. Growing up in Houston, Texas, I had been the first grandchild in my family and for years, I was used to getting my way on certain things. When my parents took me to visit my grandparents in the Fifth Ward, "just a stone's throw from Downtown Houston," my aunts spoiled me with unbridled affection and appreciation for who I was. See Figure 1a. Flashbacks of my most vivid childhood memories in Fifth Ward include the roar of Houston interstate highways, the freshly cut lawns, blight, smells of Friday afternoon barbecuing and music, drug addicted men and women walking the streets, elders on porches enjoying the cool air of the afternoon after a hot summer morning, the sizzling fry of catfish on

Friday evenings, and the gumbo on Christmas afternoon. I was a connoisseur of my aunt's gumbo and knew all the delectable ingredients: crab legs and torsos, chicken, spicy sausage, shrimp, and okra. I tried to learn this familial cultural heritage the easy way. The easy way was not going to work.

A year later, I discovered an exquisite colorful quilt that was tattered but folded neatly in the bottom drawer of the bureau in my parents' guest bedroom in their home. I asked my mother, Johnnie Paris Frazier, who made this and if I could take it with me to New Orleans. "No, you've ignored it for so many years and just because you found it again, now you want it." See Figure 1b.

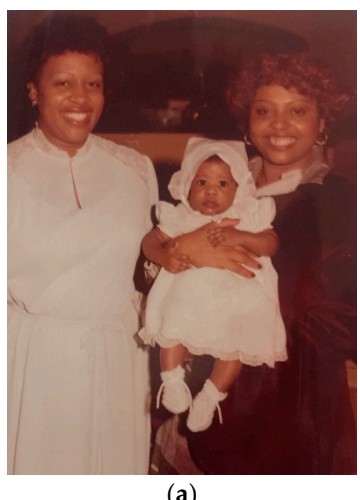
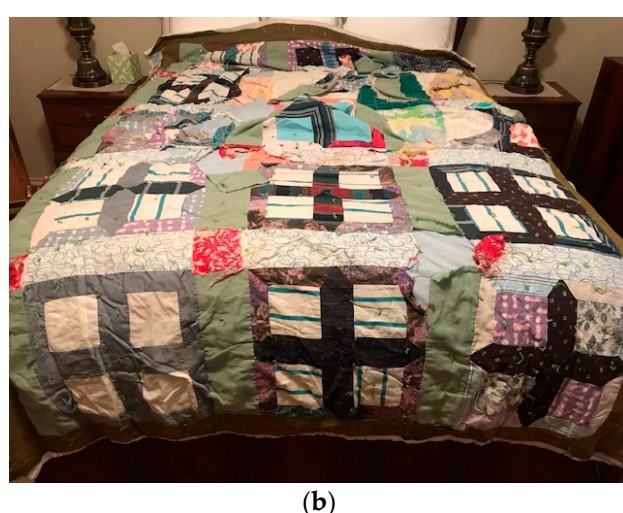

(**a**)　　　　　　　　　　　　　　　　　　　　　　　　　　　(**b**)

**Figure 1.** (**a**) My Aunt Sandra (*left*) and my Aunt Sissy (*right*) with me at my christening at Second Calvacade Church in Fifth Ward, Houston Texas; (**b**) One of my grandmother's, Annie Lou Tatum's, quilts at my parents' home in Houston, Texas.

Once again, the easy way was not going to work. Johnnie grew up in the small town of Palestine, Texas, about 150 miles north from Houston. In the summer, her mother "carted" all of her daughters back to a 240-acre farm outside of Stevens, Arkansas to spend time with her family. Johnnie explains how much work went into the quilts and how they kept her connected to her roots:

**Johnnie:** Throughout the spring and summer months, my grandmother's lady friends in the community saved up their scraps of materials in big sacks and came over to her house a few times a week in the wintertime to work on the quilts. They would do that for hours—talking, laughing, telling stories, and we would listen to them thinking we'd hear some good gossip. If they thought we were listening too close they gave us some chores to do: "Go make some butter."

**Denise:** Was that a real request, by the way?

**Johnnie:** Oh, gosh, yeah! We had to churn our own milk, make our own butter when we went to visit my grandmother. I used to hate it. We picked peas, we pulled corn, we killed chickens to cook the afternoon dinner. Oh, Lord! What we didn't do? During the winter, they would go in and do the quilting. My grandmother had what we called a "quilt closet." When my sisters and I were kids, whenever we got cold, we just opened up this closet and grabbed a quilt. The quilts were just stacked all the way up and sometimes we pulled at a quilt in the middle and they all tumbled out. "Oh my God!" Our grandmother took pride in folding her quilts and we knew we had to fold those quilts back up right. But when they would come out, I was like, "Oh, look at them!" The patterns of some of these quilts were just so beautiful. She gave all her grandchildren a quilt. That was her gift to us.

Johnnie took this quilt with her when she moved to Houston to attend Texas Southern University. As I came to terms with my mother's decision to withhold it from me, I realized that my decisions for

moving away from Texas for the purposes of "self-discovery" and graduate education had taken me further away from what I am now researching; my family history. In the years that I have been away, I have become an "Ex-Tex" and my family had no problem letting me know it.

Still, the land, the water, geography, foodways of Houston—the warm hugs, my father's mother's smile at the kitchen table, her husband's cut off finger and rough hands tightly gripping mine—have defined who I am and what I believe. They make up the cityscape of my youth—a term which geographer Richard Campanella defines as "the urban corollary of a cultural landscape, the *tout ensemble* of all that constitutes a city: its underlying urban landscape, its biota and water, its built environment fashioned by various human agents and their settlement patterns, its infrastructure and idiosyncrasies" (Campanella 2017, p. xiii). The cityscape of Fifth Ward has motivated me to honor the memories of my ancestors, their life trajectories, families, and environment. In a city obsessed with rapid change and development, the neighborhood remains a cultural anomaly.

In Fifth Ward, my aunts and father still call each other by their childhood nicknames. My Aunt Wanda Darlene Hill is "Sissy." Sandra is affectionately called "Sanra," and my father, Winfred Frazier's, nickname is "Jitterbug"—shortened to "Jitter" and more accurately pronounced "Jittah." I always thought his name appropriately addressed the spirit of my father—a pursuer, a restless wanderer, a mover who was forged by the Fifth Ward streets, the African American women who educated him with care and intelligence, and the hustlers who taught him to trick the tricksters. My father told me, "Growing up in Fifth Ward defined me as a person later. I felt like I could handle the most difficult circumstances. Even as an early child, I was able to deal with difficulties—you have to understand what it takes to be patient."

While growing up in southwest Houston, I contemplated the habitus of Fifth Ward in the way my father swaggered his way through the hallways of Edgar Odell Lovett Elementary School; a school in a suburban neighborhood very distant from that of his upbringing in Fifth War (Bourdieu 1987). He always responded the same way to my elementary school teachers who asked him how he was doing: "Not too bad for an older guy." The familiar refrain somehow got translated into my child's brain as "Not too bad for a lonely guy." For many years, I questioned and contemplated the loneliness of my father in suburbia—the uncomfortable laughter of the teachers as he finished this awkward joke. I wrote this article to bridge some of the gaps I felt in these hallways. Situating my family narrative alongside African American history in Texas through storytelling, oral histories with my parents, and archival images, I hope to spark the interest of every other person who has felt just out of reach of their own sense of home and history.

## 2. African American History in Texas

Houston's population is 44.8% "Hispanic." Further, 22.5% of its population identifies as African American (U.S. Census 2018). Yet, when I was growing up, I did not understand how we all got here. As African descended people of the South, our personal histories often clash with "official" Texas histories which avoid discussing how slavery in Texas, an institution of white terrorism and horror, is central to the founding of the state (Beeth and Wintz 1992). Part of how my family resisted this white washing of history was to not say very much at all, but to work to have a better life for their children. See Figure 2.

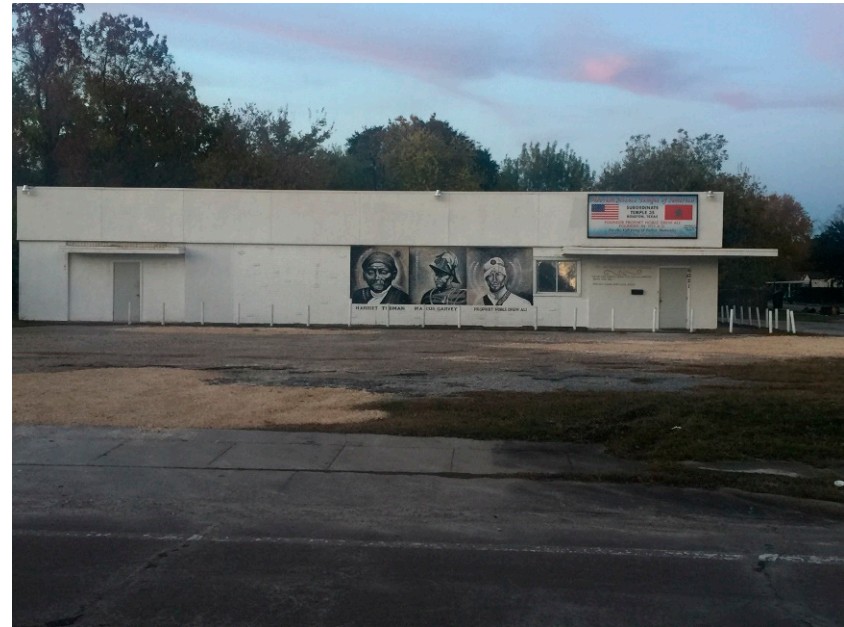

**Figure 2.** African American history on the side of the Moorish Science Temple of America, across the street from Second Cavalcade Missionary Baptist Church in Fifth Ward, Houston. The portraits, from left to right, include underground railroad conductor Harriet Tubman, Pan-Africanist Marcus Garvey, and founder of the Moorish Science Temple of America, Prophet Noble Drew Ali. Photograph by Denise Frazier.

The oral histories I conducted with my parents finally gave me an excuse to talk directly to them about their pasts and chronicle their experiences. When I asked them about our family's history in Texas, Winfred, generally tight-lipped, simply said, "things were difficult" for his parents, and "You know, sometimes the things that you go through as an individual have long-term effects on you. And sometimes, they never want to talk a lot about it." As I asked more questions, he elaborated:

> Texas had a lot of slavery. And when we think of slavery, I do not just mean people who owned other people. I mean the mentality, the lack of being treated as equal. It still exists. Abraham Lincoln signed the Emancipation Proclamation to liberate the slaves in 1863, but Texas was so far West, people enslaved in Texas didn't learn they were free until June of 1865. Growing up, we knew about Juneteenth. It wasn't talked about much in school. In fact, African American history really wasn't taught much in the Houston Independent School District at all.

My parents have spent years volunteering in the public schools in Houston to try to be role models for African American youth and to support my brother and me with their advocacy and visibility. Johnnie said that one of the things she came to realize as a parent, and also an educational advocate, is:

> American history books were controlled by a white conglomerate committee, so they never thought black history was important to teach, and they still don't. I mean, they may mention one person—Dr. Martin Luther King—in one sentence or paragraph, but you're not going to know anything else. In fact, and if you did not take it upon yourself to educate your children on black history—that there was a black man who invented the clock, or whatever—you wouldn't know anything about your own history.

Johnnie's understanding of how state policy decisions shape the way that young children learn history is explored in depth in James W. Loewen's decade-long study of U.S. history textbooks. In *Lies My Teacher Told Me: Everything Your American History Textbook Got Wrong*, he explains:

Publisher pressure drives in part from textbook adoption boards and committees in states and school districts. These are subject in turn to pressure from organized groups and individuals who appear before them. Perhaps the most robust such lobby is Educational Research Analysts. (Loewen 1995, p. 211)

Over fifty years old, the Educational Research Analysts were run by the husband–wife team of Mel and Norma Gabler. On their website, they share more of their background:

We are a conservative Christian organization that reviews public school textbooks submitted for adoption in Texas. Our reviews have national relevance because Texas state-adopts textbooks and buys so many that publishers write them to Texas standards and sell them across the country.

Amongst their stated "subjects of concern" are "original intent of the U.S. constitution," "respect for Judeo-Christian morals," and "politically-correct degradation of academics." Textbooks that the organization has approved taught my seventh-grade class about "Texas heroes" such as Davy Crockett and James Bowie who fought at the Alamo. The unit inspired immediate boredom and ambivalence to my 12-year-old self. They were all white men in colonial gear who were killed trying to kick the Mexicans out of their own country. Of those lessons on the last days of Mexico's control of Tejas, I took away only a few things. One: Antonio de Padua María Severino López de Santa Anna y Pérez de Lebrón (also known as Santa Anna) had a cool name. Two: Images of Santa Anna dressed in fierce and flashy military wear interested me. See Figure 3.

Three: the Alamo (period). And lastly, a question: Where were Native Americans, the African-descended people, and the women (of various races, ethnicities, and nations) in Texas history?

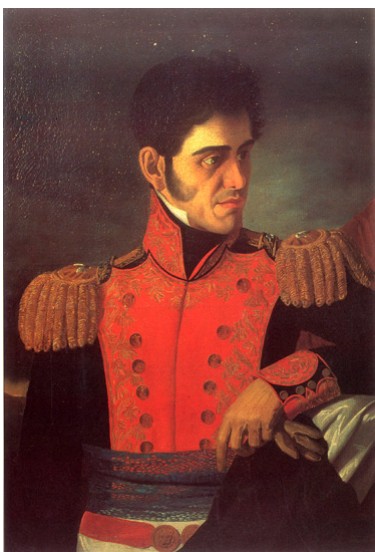

**Figure 3.** Fierce and Flashy: A Painting of Antonio López de Santa Anna, courtesy of Museo Nacional de Historia, México.

It was only in adulthood that I realized that my boredom was indicative of my quiet resistance to what I was learning. The history books had obscured the fact that the Battle of the Alamo was a defining moment in Anglo settlers' deep allegiance to a system of race-based discriminatory practices that upheld a slave economy. For instance, one of the "defenders" of the Alamo who was killed by Santa Anna's forces, James Bowie, built a career slave-smuggling with the pirate Jean Lafitte in Louisiana (Davis 1998, p. 61). In retribution, Sam Houston led U.S. troops in a surprise attack of Santa Ana's forces at San Jacinto. Historian Carlos Casteñeda writes of witnesses clubbing and stabbing Mexican soldiers, some on their knees. After the battle, Santa Anna was forced to cede Texas and

Houston declared it an independent republic for the "Anglo race" (Takaki 1993, p. 174). Yet, the history of Houston's life included a great deal of personal ambivalence to the racial hierarchy that he was often hired to uphold (Haley 2002). As a teenager, Houston left his family's home in Tennessee and lived with a Cherokee community on Hiwasee Island and was adopted by the chief, Ahuludegi. He learned the Cherokee language and took on the name Ka'lanu, which translates as Raven. In 1817, he found work with the federal government's removal of the tribe from the state and joined Cherokees who fought against the Muskogee in the Battle of Horseshoe Bend, where 800 Muskogee (Creek) were slaughtered (Dunbar-Ortiz 2014, p. 99). A dispute with Secretary of War John C. Calhoun developed when he showed up for a meeting with Cherokee leaders wearing Cherokee clothes, and he resigned from his job and took up politics.

As the governor of Tennessee, he married into a wealthy slaveholding family, but when his marriage fell apart, he returned to his adopted Cherokee family. They gave him tribal membership and asked him to negotiate with Andrew Jackson's government who was implementing the Indian Removal Act. He was eventually able to help them secure land in Arkansas Territory (Haley 2002). In school, we were not taught these previous lives of Sam Houston, nor were we taught the history of indigenous displacement in Texas. To tell the truth about our history is to gift future generations with a greater sense of empathy and provide them with the tools they need to combat white supremacy, nationalism, and persistent inequalities.

## 3. The Whitening of Texas

In the 17th and 18th century, the area known as *Tejas*, claimed by the Spanish empire as part of "New Spain," was home to indigenous polities who traded and traversed geographical borders known to each other. Tribes in eastern Texas included the Atakapa-Ishak, whose territory reached into Louisiana from Galveston Bay and Akokisa, who lived within the territory that is present-day Houston (Barr 2011, p. 15). Further west, indigenous resistance to colonization by the Apache and Comanche is one of the main reasons the area remained scarcely populated by Spain through the Mexican Revolution (Barr 2011, p. 40). In 1821, when Mexico won its independence from Spain after a decade of fighting, only 3500 settlers lived in Tejas amongst many indigenous nations (Weber 1973). Around the same time, settlers from the United States began crossing the newly created state of Louisiana:

> Many of them were slave owners from the South in search of new lands for cotton cultivation. President John Quincy Adam tried to purchase Texas for a million dollars in 1826, but Mexico refused the offer. (Takaki 1993, p. 173)

In 1829, the Guerrero decree outlawed slavery in Mexico and prohibited further immigration of American settlers into Tejas, which sparked Anglo slave-owners to rebel against the government. Anglo settlers continued to bring enslaved Africans into the department and they continued to cross the border illegally. By 1835, they outnumbered Mexican inhabitants 1/5—20,000 to 4000. Stephen Austin encouraged settlers to come, claiming that Mexicans were a "mongrel Spanish-Indian-negro race" and that the white American slaveholders represented civilization (Takaki 1993, p. 173). After Mexico ceded Texas in 1836, the Republic of Texas' newly established legislature ordered all free black people out of the Republic (Loewen 1995, p. 53). They also disputed the southern border, which has been drawn at the Nueces river. In 1846, the U.S. occupied the area between this river and the Rio Grande, which escalated into war. The *intervención estadounidense en México* (U.S. Intervention in Mexico), or the Mexican American War, as it is known in the United States, led to the U.S. conquest of Alta California and Santa Fe de Nuevo Mexico—present day California, New Mexico, Nevada, as well as parts of Colorado, Arizona, and Utah (Takaki 1993, p. 176).

In 1845, Texas came into the union as a slave state (Campbell 1989). In the years leading up to the American Civil War, cotton and sugar plantations developed in East Texas and Houston was one of the epicenters of trade in the region. Yet, the state's leadership in Congress did not agree with the continued expansion of slavery. Again, the life of Sam Houston shows the contradictions of the

institution within one man—a slave-owner, he taught the people he held in bondage to read. As a U.S. senator, he opposed the 1854 Nebraska-Kansas Act, which allowed states to determine whether they would come in as free or slave. As governor of Texas, he refused to pay bounty for the return of runaway slaves and eventually opposed Texas's decision to join the Confederacy, which led to his removal from office (Haley 2002).

## 4. Emancipation?

In telling the story of African American history in Texas, many people, like my father, focus on the end of slavery altogether. On 1 January 1863, Abraham Lincoln's Emancipation Proclamation went into effect: Enslaved Africans in states that had succeeded from the Union were said to be free. Far from the heart of the Union Army, 250,000 enslaved people in Texas knew nothing of the order. In fact, when the Confederacy surrendered in April of 1865, Texas did not officially rejoin the Union until 19 June 1865 when 2000 Union soldiers arrived in Galveston (Moskin 2004). On this day, General Gordon Granger (Granger 1865) read General Order Number 3:

> The people of Texas are informed that, in accordance with a proclamation from the Executive of the United States, all slaves are free. This involves an absolute equality of personal rights and rights of property between former masters and slaves, and the connection heretofore existing between them becomes that between employer and hired labor. The freedmen are advised to remain quietly at their present homes and work for wages. They are informed that they will not be allowed to collect at military posts and that they will not be supported in idleness either there or elsewhere. (Granger 1865)

This momentous day became known as Juneteenth, a holiday that African Americans in Texas, and now people around the country, celebrate (Moskin 2004). Yet the language of the order to "remain quietly in their present homes" was ominous as well. In the years after, Texas, like other former Confederate states:

> enacted sweeping vagrancy and labor contract laws, supplanted by anti-enticement measures punishing anyone offering high wages to an employee already under contract . . . Louisiana and Texas, seeking to counteract the withdrawal of Black women from field labor, mandated that all contracts shall embrace the labor of all members of the family able to work. (Foner 1988, p. 200)

The Freedmen's Bureau, which was founded by the federal government to help formerly enslaved people, documented the white backlash to freedom. Between 1865–1868, the Bureau documented over 1500 acts of violence, including 370 homicides, by white Texans against African Americans (Crouch 1992). Susan Merritt, a freedwoman from Rusk County, Texas remembered seeing black bodies floating down the Sabine River (Foner 1988, p. 119). In 1866, U.S. Congress passed the Civil Rights Act, which established "Reconstruction" in the South (Smallwood 1981). Like the early history of Texas, the history of this timeframe has largely been told by white Southerners who opposed equal rights for freed people. The power in reclaiming this time in our history allows us to imagine other futures.

During Reconstruction, Texans elected Edmund J. Davis, a Galveston judge who fled the rise of the Confederacy and joined the Union Army as governor. Under his leadership, he took on the rise of the Ku Klux Klan by organizing a State Police force of 200 members; 40% were black:

> Between 1870–1872, the police made over 6000 arrests, effectively suppressing the Klan and providing freedmen with a real measure of protection in a state notorious for widespread violence. (Foner 1988, p. 440)

Without the fear of retribution, African Americans began to register to vote and buy property. In Houston, they populated the meagerly settled strip of north Houston land. Many of these

communities were composed of ship industry workers, newly freed African descended people, and former residents of Louisiana and East Texas. See Figure 4.

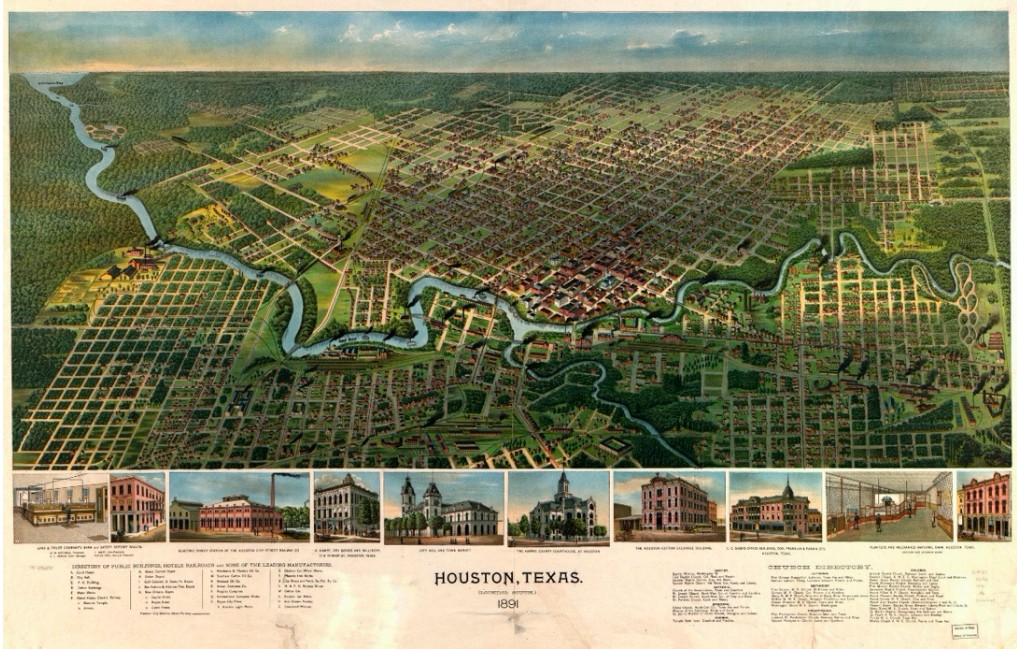

**Figure 4.** A map of Houston, Texas from 1891, courtesy of the Library of Congress. Despite high elevation (in Fifth Ward, 43 feet above sea level), Houston's topography is flat. On this map, you can see Galveston Bay and Buffalo Bayou and White Oak Bayou flowed through the city. They continue to articulate the borders of neighborhoods. Some streets are paved with the familiar concrete, with street drainage ditches to prevent excessive rainwater from entering homes and flooding roads. The depth of the ditches and the large pipes fascinated me as a child.

A group of ministers raised money through their churches to buy a 10-acre lot of land in Third Ward, which they named Emancipation Park (Hardy 2015). Black representatives were elected to the city council. By the early 1890s, 100,000 African Americans voted in Texas elections (Foner 1988). See Figure 5.

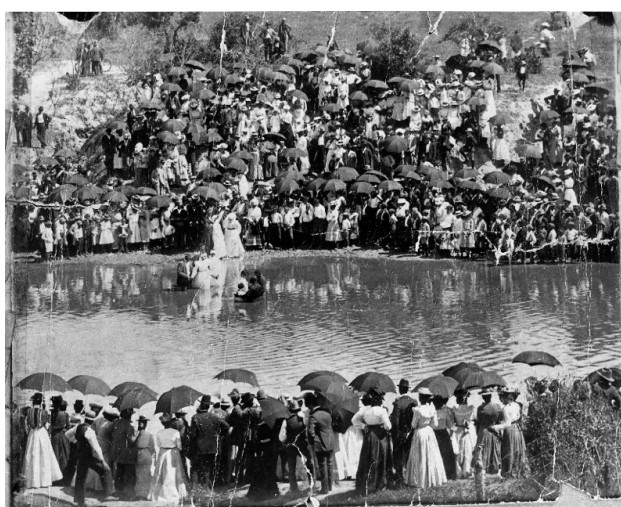

**Figure 5.** African Americans in Texas gather to witness a baptism in Buffalo Bayou near Houston, Texas in 1990. Photograph courtesy of Special Collections, University of Houston Libraries.

However, the Supreme Court's decision in *Plessy V Ferguson* 1896, which legalized segregation in the United States, led to systematic disenfranchisement of people of color once again. Church on Sundays became one of the only times when black communities were able to congregate in large numbers and Juneteenth began to be more than just a celebration of the past. See Figure 6.

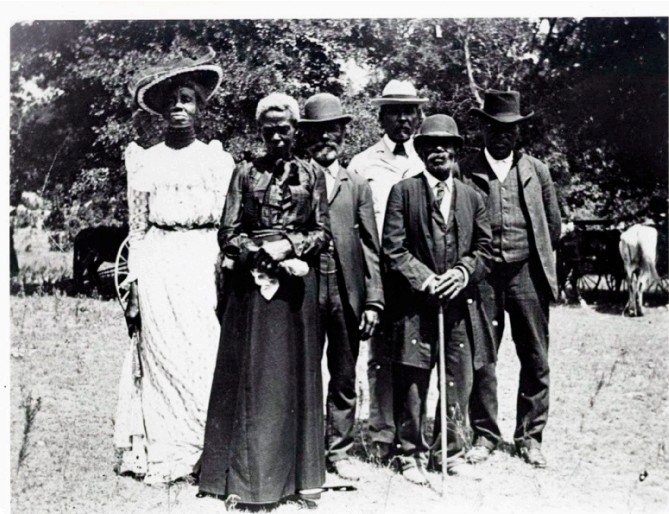

**Figure 6.** Emancipation Celebration on 19 June 1900. Image from the Austin History Center, Austin Public Library, courtesy of Mrs. Charles Stephenson (Gracy Murray).

In 1902, the state legislature in Texas created a poll tax and a year later, implemented "white primaries that dramatically reduced the voting numbers of African descended Texans to 5000." Public schools and other spaces became segregated. (Smallwood 1981). My grandparents were born the generation afterwards. My father's mother, Emmola Taylor Frazier, was born outside of Sweet Union in Lufkin County. In an article for *The Texas Observer*, Jake Bernstein writes about the small town that she came from:

> The community of Sweet Union in Deep East Texas is not on highway maps. It's easy to drive past it and see only the soft undulation of the land and the cattle grazing in the meadows, missing the dilapidated shacks set back from the road. (Bernstein 2002)

This landscape is connected to the failures of Reconstruction. My father explains what his maternal grandfather, D.H. Taylor, said about growing up in the area:

> He was a farmer. As a young man, he was a deacon in the church. He told me when he got on his knees to pray to God, he always made sure his knife was always open in his pocket. You never felt safe anywhere—whether it's from other people who lived in the city or police game wardens who didn't want you to hunt. It was at the height of racism, discrimination and Jim Crow segregation. He eventually came to Houston. They saw the Houston area as a better place to at least try to get a better life. And that's how they ended up in Fifth Ward in the late 1940s.

My father's paternal family migrated to the same neighborhood from Sibley, a small town in northeast Louisiana. He reflects on the silences in his own father's life:

> My father's father, Albert Frazier, Sr., never did leave Louisiana. He primarily raised vegetables, but also some cattle. My father was a man of very few words. He did take us back to visit the farm around Sibley during the holidays or on summers, but I didn't spend a lot of time there. Most of my father's immediate family members—his brothers and sisters—left to come to Houston, too.

The city was created around bayous that ran alongside strips of land. The Fifth Ward of Houston is composed of the cession of two other wards, First Ward (east of White Oak Bayou and Little White Oak Bayou) and Second Ward (north of Buffalo Bayou). In the 19th century, the ward's residents consisted of Jewish, Irish, Russian, and Eastern European immigrants escaping famine or religious persecution, in the case of Jewish immigrants (West 1979, p. 95). On 21 February 1912, a large fire burned a great deal of the neighborhood. Known as the "Great Fifth Ward Fire," in the aftermath of the damage, many African Americans from Louisiana and East Texas began to migrate to the ward (Pruitt 2012). See Figure 7.

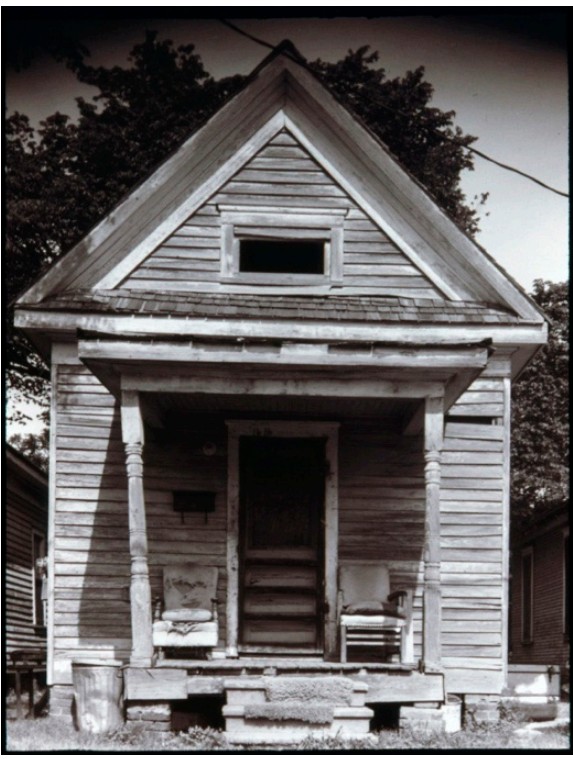

**Figure 7.** Shotgun in Fifth Ward, Houston from the 1970s. Photograph by Danny Lyon.

In *Houston Bound: Culture and Color in a Jim Crow City*, Tyina Steptoe writes:

Once a sleepy town on the banks of Buffalo Bayou, interwar Houston was a bustling city with little memory of its former status as a frontier community on the western edge of Dixie. The railroad yards and ship channel docks held the promise of wage labors for farmers and sharecroppers, and thousands responded. They headed southwest to Houston using the red, dirt roads of East Texas; they travelled westward from the flood-ravaged swamps of lower Louisiana. (Steptoe 2016, pp. 61–62)

After the Mississippi River flood of 1927, many Afro-Creole families from southwest Louisiana migrated to the neighborhood as well. Their neighborhood on the northern edge of Fifth Ward was named Frenchtown. In "Frenchtown, Texas," Nathan Rivet's shares more about this part of Fifth Ward:

Frenchtown was mainly French-speaking and in 1930 had a population of approximately five hundred Creoles. The community members were largely Catholic, and their life revolved around Our Mother of Mercy Roman Catholic Church. The Creole identity was present in many fields: residents had a distinct language, religion, cuisine, and music. Frenchtown also distinguished itself by its colorful *patios*, its cuisine, and its music: zydeco. That music, a blend of traditional Creole music and Houston's blues and R&B, would eventually spread from Houston across African American communities throughout the West. (Rivet 2017)

Music clubs like Mr. A's and the Continental Zydeco Ballroom hosted dances. Clifton Chenier played with Lightnin Hopkins at the Silver Slipper (Wood 2003). Of the 63,337 Houstonians marked as "Negro" in the 1930 census, around 19% were born in Louisiana (Steptoe 2016, p. 63). In the 1980s, remnants of this community had mostly faded, but could still be felt in the sounds of two-steps playing on crackling radios outside homes or the steamy smells of hot seafood gumbo on a cold Christmas day.

## 5. Raising a Family in Fifth Ward, Houston

Post-World War II immigration, which brought my own family to Houston, was partly driven by the steel industry's connections to the Port of Houston and the Houston Ship Channel, a waterway that was created to permit the sea vessels to travel between the Gulf of Mexico and terminals in the city. The ARMCO Steel Corporation, for instance, was strategically placed 12 miles from the Houston Ship Channel and employed many African American workers, including my grandfather (Rankin 2017). My father, Winfred, explains:

> At the time, it was one of the nation's larger steel corporations, and he worked there for about 40 years. So that was a good job and that opportunity would not have existed for him back in Louisiana.

With stable incomes, black-owned businesses developed in Houston's Third and Fifth Wards (Pando 2002). Winfred's aunt, for instance, ran a restaurant in an area called Liberty Gardens, just north of Fifth Ward. Winfred says:

> My mother's sister, Batts, had a little restaurant with a juke box. She served good food, so people went there to eat as well. They could order seafood and hamburgers and were served beer. A lot of her clients were longshoreman because her husband was a longshoreman. A lot of the guys he worked with on Eastern Ship Channel would go to her place to hang out. They all knew each other.

On the weekends, the commercial corridor of Lyons Avenue supported a music scene with black-owned hotels and lounges catering to musicians on the "chitlin circuit." Movie theaters, eateries, and shops served to strengthen the African American economic wealth and solidified Fifth Ward as a place where families could grow, be entertained, and thrive. See Figure 8.

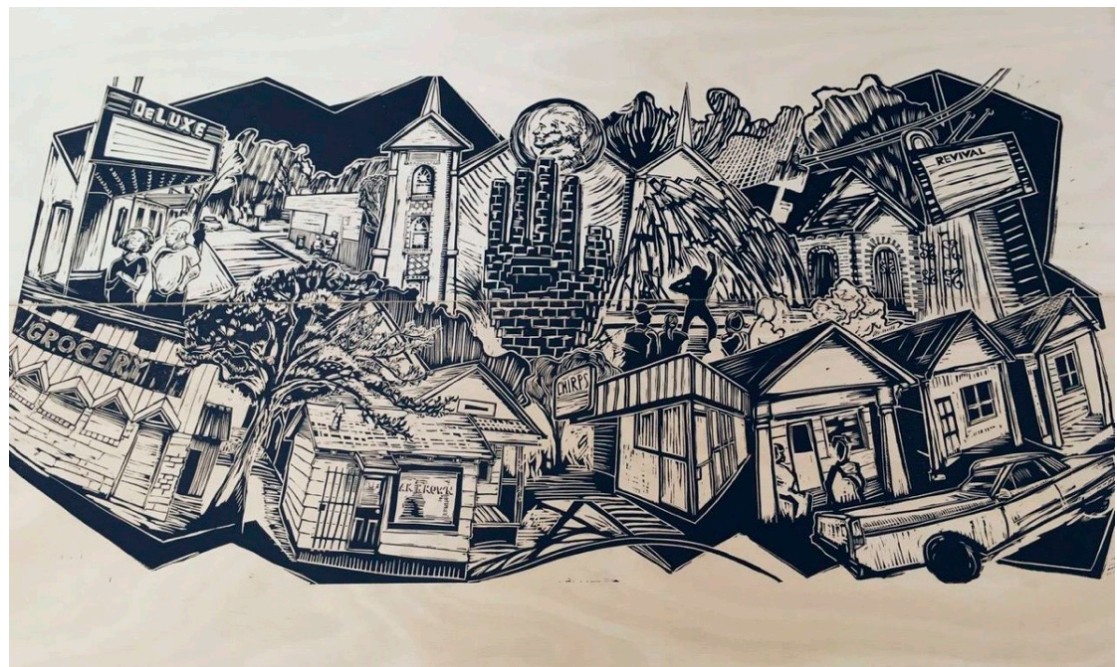

**Figure 8.** Siver Black's print installation, "Nickel Views," includes multiple hand drawn images of the Fifth Ward. Photograph by Emily Sloan, courtesy of Mystic Lyon.

Diana Kleiner recounts some of the development in the community:

Peacock Records, a recording company founded by music entrepreneur Don Robey and named after his popular Bronze Peacock Club, started in the ward, as did C. F. Smith Electric Company, one of the state's early licensed electrical-contracting companies. Finnigan Park, the second public park for blacks in Houston, opened in the community in the postwar years, and the Julia C. Hester, a black community center, began service. (Kleiner 2010)

Don Robey later became the founder of Duke Records, a Fifth Ward-based record label that achieved significant success with Tennessee-born artist Bobby "Blue" Bland. In 1957, his song "Further up the Road" reached Number One on the Billboard charts and is still considered a Texas blues staple. Other artists who recorded with Duke Records include Junior Parker, Big Mama Thornton, and Johnny Ace (Wood 2003).

My grandparents, Alfred Frazier, Jr. and Emmola Taylor, met each other during this era. A graduate of Phillis Wheatly High School, Emmola was a nurse assistant; a position that placed her alongside other African American women who made contributions to the medical industry with their professional training, care, and labor. Many of these women paved the road for future generations, like my mother, to enter the medical profession as physicians. With steady incomes, they started their family in a three-bedroom Fifth Ward home in Kashmere Gardens. See Figure 9a–c.

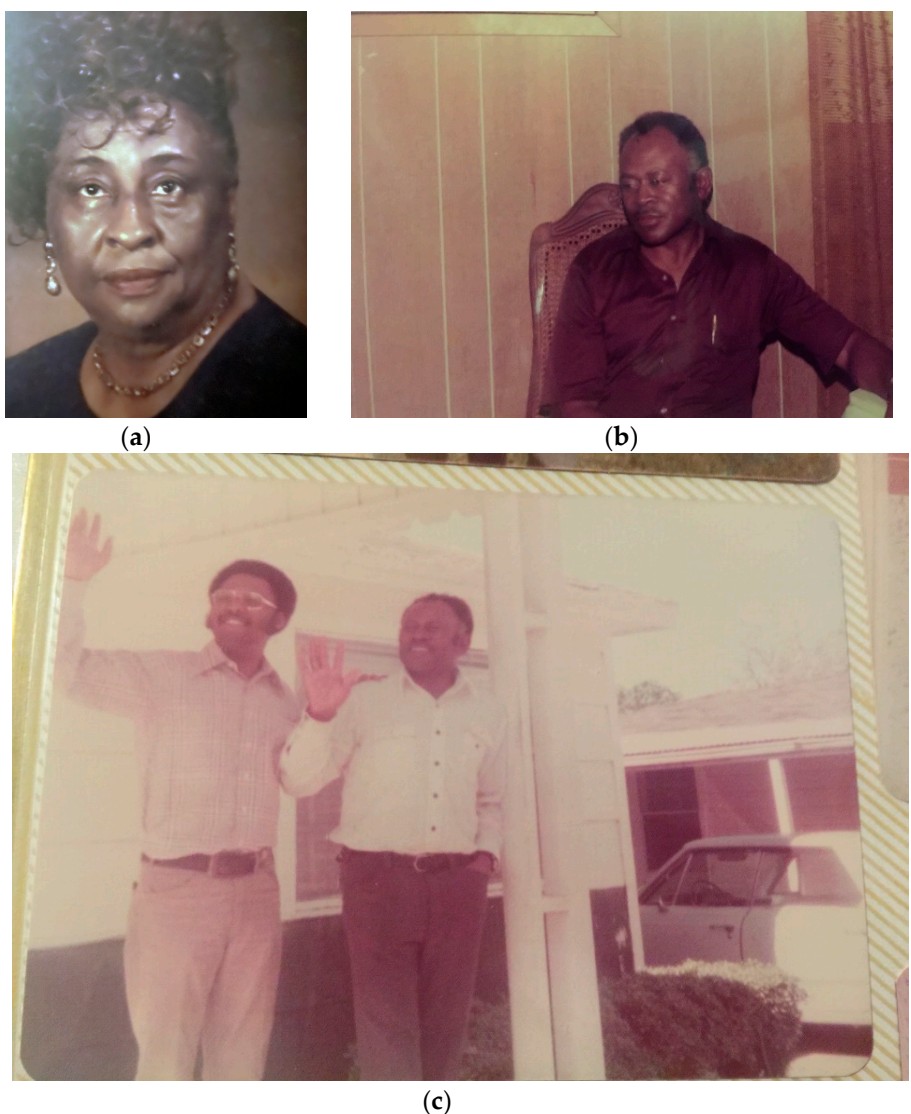

**Figure 9.** (**a**) Emmola Taylor Frazier; (**b**) Albert Frazier, Jr.; (**c**) Winfred Frazier with Albert Frazier, Jr. in front of their Fifth Ward home. Photographs courtesy of the Frazier family.

When I asked my dad about his childhood home, my father shared stories that blended the city and country together in their family rituals:

Our home was not a fancy home, but we did have lots of space. We had a big backyard. My father kept a really great garden back there. Family members came over to get summer vegetables—he grew tomatoes, okra for gumbo. My father was also very good at barbecue. He had a big barbecue pit in the backyard and cooked for the whole family. Some of the briskets could take hours.

The family house was a safe place within turbulent times. Although *Brown V Board of Education* ended de jure segregation in 1954, Houston's public-school system remained segregated until 1984 (Bryant 2004). My father took the public bus to an elementary school outside of Kashmere Gardens on his own every day. As the older male child of the family, he was given more license, which led to many occasions where he was directly confronted with the rough streets of Fifth Ward. He recalls:

I had to catch the bus on Lyons Avenue to get to school. I'd have to go through the gauntlet of all these hoodlums on the corners to make it to the bus stop. You know, the differences in

education today is that kids don't generally think about worrying how they are going to get home. For me, when I look up and the clock would be 2:45, and I know the bell's going to ring at three. I'm going to have to walk to Lyons Avenue and get to that bus stop. It was traumatic. The only thing I had was bus money. I started hiding my money in my socks because they would shake you down. You didn't want to cross these corners.

Harmonica player and singer Juke Boy Bonner, the "Bard of the Fifth Ward," wrote songs that issued warning to "Stay off Lyons Avenue" or it could be "the last place you'll be seen". In another song from 1969, he sings about "Struggle Here in Houston":

There's some streets in Houston
I stay clear of after dark
Cause some cats that'll bump you off
Just to hear their pistol bark.

In high school, my father attended his mother's alma mater, Phillis Wheatley, while my aunts attended Kashmere Gardens. See Figures 10–12. He fondly recounts being taught by talented African American teachers who kept him on track for college:

These were women who, because of discrimination, did not have the options of their white-counterparts, other than teaching. The female teachers were brilliant.

Public schools began desegregating at an all-white elementary school in Kashmere Gardens in 1960, but water fountains, bathrooms, and the cafeteria remained segregated inside the school. Only a few children were admitted (Bryant 2004). By the mid-1960s, desegregation of the public schools was at the heart of the civil rights movement in Houston:

In 1965, more than 9000 blacks boycotted schools in support of integration and about 2000 blacks marched 62 blocks to HISD headquarters, where they were met by barbed wire, padlocked fences, and police. Five years later, 3500 Hispanic students started a three-week boycott of classes in protest of a plan that paired mostly predominantly Hispanic and black schools. (Bryant 2004)

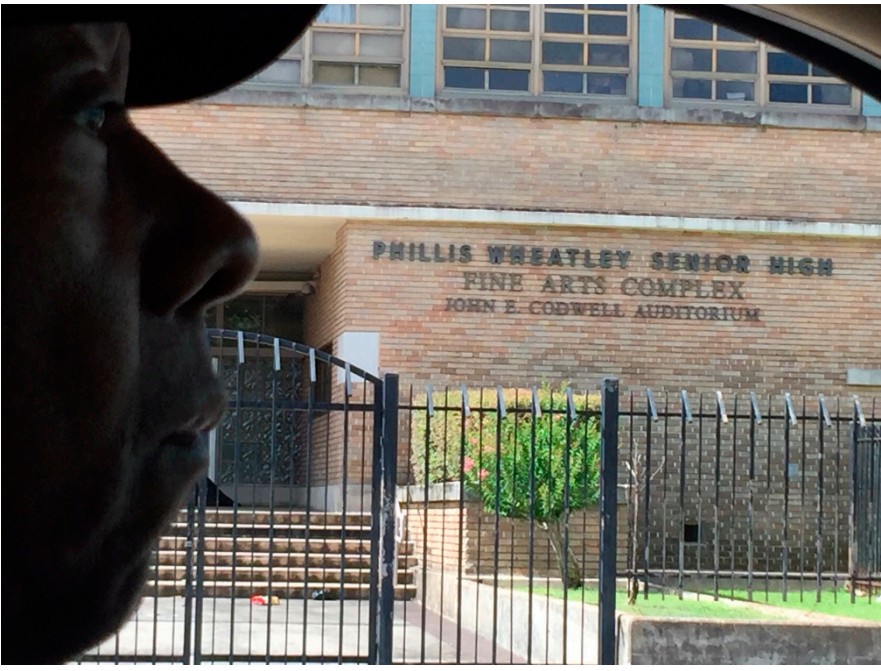

**Figure 10.** Winfred Frazier and I drive past Phillis Wheatley Senior High School in 2018. Photograph by Denise Frazier.

Coming of age at Phillis Wheatly during the Civil Rights Movement influenced a number of my father's peers to go into politics. Mickey Leland and Barbara Jordan served in Congress, Carl Hampton became a leader of the People's Party, and brothers El Franco Lee and Bob Lee had long-term commitments to Houston and economic justice organizing. El Franco was the first African American commissioner for Harris County Precinct 1, while his brother, Bob Lee, stayed closer to the grassroots as a community organizer and unofficial "Mayor of Fifth Ward." The latter left Houston for many years, helping to organize multiracial political organizations. At the 1968 Democratic National Convention in Chicago, he was filmed reaching out to the Young Patriots, an organization of young white residents of the city:

> We come here with our hearts open . . . Once you realize that you are paying taxes-taxes-for the cops to whoop your ass. . . . You're paying them to come in to beat your children. You're paying them to run you off the corners and you're paying them to kill you and deal from there. The same thing is happening on the south side and the west side. And when you can realize that concept of poverty—a revolution can begin. (George 2017)

Back in Fifth Ward, the Mayor of Fifth Ward was attuned to the way that race was used to undermine working class solidarity and connected to struggles of people around the world. See Figure 11.

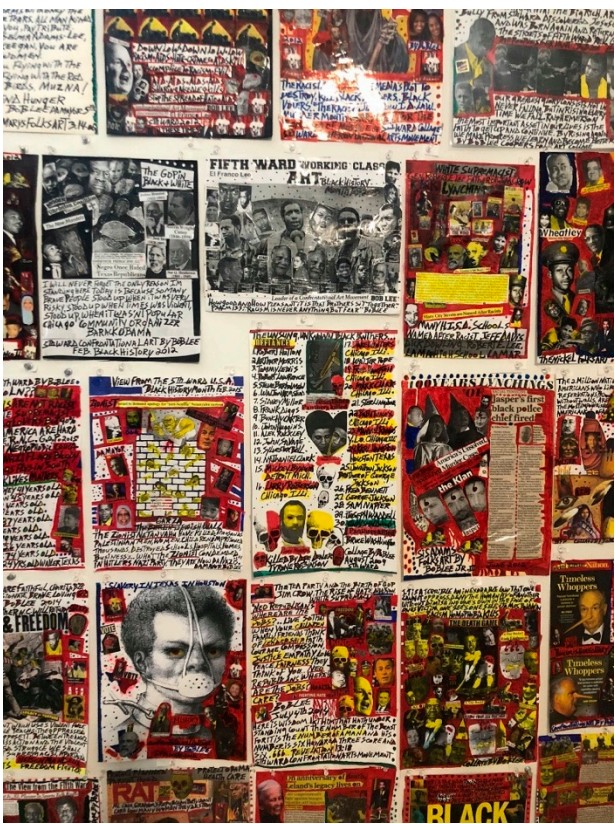

**Figure 11.** Collages of Black resistance around the world and in Fifth Ward by Bob Lee. Image courtesy of Mystic Lyon.

Poverty had become more pervasive as former Fifth Ward residents relocated to other parts of Houston and desegregation left many black-owned businesses behind as African Americans shopped at white establishments, but white shoppers did not reciprocate:

> As white-owned stores welcomed their new black customers, Nickel retailers lost out. At the same time, new neighborhoods and racial integration of existing ones attracted

many younger, more prosperous [African American] families [to other parts of Houston]. Meanwhile, freeways came; I-10 and U.S. Highway 59, in the words of historian Joe R. Feagin, "literally crucified the area by creating large freeways in a cross pattern through its heart". (Pando 2002)

Famous artists such as blues musician Weldon "Juke Boy" Bonner navigated the challenges of living and playing music in an area that faced such drastic economic change in a short period of time. Lorenzo Thomas' *Juke Boy Bonner Sang the Blues* recounts the difficulties he faced in the economic transition of a relatively prosperous Fifth Ward music scene in the 1950s to the economic deterioration of the 1970s:

> Back in Houston, Bonner struggled to make a living with his music. But he was caught with a major change in the music business and the city itself. The major recording companies were busily buying out or bankrupting the smaller rhythm and blues labels, creating new stars and slowing putting an end to the black touring circuit. Lyons Avenue, through a combination of overcrowding, absentee landlords, and bank 'redlining', was becoming a less glittering street while the Fifth Ward itself became more and more of a depressed area. (Thomas 1979)

As businesses began to close, food insecurity plagued the district. There were fewer grocery stores and the ones in the neighborhood had higher prices, which took advantage of residents who lacked viable transportation options. Many of my dad's childhood friends have died or deteriorated due to drug-related causes. Throughout these changes, churches in Fifth Ward were community anchors. There was one, my father says, "on every other corner." My grandparents were active members of Second Cavalcade Missionary Baptist Church on Liberty Road, where my grandfather was later granted the position of trustee and deacon. Winfred explains that his father found leadership opportunities as well as spiritual guidance at his church:

> My grandfather was one of the deacons of the Second Cavalcade Missionary Baptist Church. He joined the congregation right after the church was founded. Churches were not only the central point of being a religious person, but also a meeting place for people to go at least one day a week to get together and not have to face and deal with the pressures of what the other six days were like.

> Growing up, we had the same pastor for maybe 25 years. He and his wife started the church and stayed there for a long time. They were really down to earth. They didn't try to get as much money as they can from the parishioners. They asked about your tides and donation, but they didn't make every sermon about what money you need to get on this Sunday. And preached from the Bible—what the Bible teaches. It was more preserved than some of the other churches with traditional Baptist hymn music with an organist or pianist—we didn't have a drummer. I sang in the choir, was on the junior usher board, and went to Vacation Bible school there.

In the 1970s and 1980s, new suburban areas appealed to middle-class black families. Missouri City, for instance, was one the many Houston subdivision "bedroom community" towns where African American middle-class in surrounding Houston resettled. Attracted to the low property rates and newly developed properties and greenspaces, the "post-civil rights" suburb had escaped the pattern of "redlining" that was prevalent in the early to mid 20th century. Urban Planning and Geographical Studies professor Deirdre Pfeiffer describes this phenomenon:

> Families of color were more likely to own a home and less likely to be in poverty and they lived in neighborhoods with higher educational attainment on average in so-called post-civil rights suburbs when compared to central city areas and older suburbs. (Binkovitz 2015)

Due to the flight of middle-class Fifth Ward residents to Houston's growing suburban communities and cities, the neighborhood became economically impoverished. Many of the new Missouri City residents were first-time home buyers and enjoyed the sprawling accommodations that were in close proximity to Houston's growing medical center. Although they may not have received the same quality of education as white suburban neighborhoods, on average, the education and wealth gaps were smaller in these communities than for African Americans and Latinos who lived in the inner-city of Houston (Binkovitz 2015).

## 6. Challenging Black Identity in Houston

My father's story follows this migration out of the Fifth Ward into the suburban metropolis of the southwest section of Houston. After high school, he attended Texas Southern University, a predominantly black institution in Third Ward, Houston. It was founded in the late 1940s by the Texas legislature because, as my father explains, "they did not want African Americans to go to the University of Texas." The university became an important cornerstone of Black education in Houston and students came from around the state. My mother had grown up in a small town in Palestine, Texas and was "overjoyed" to be in the big city. While my dad lived at home in Fifth Ward, my mother stayed in the scholarship dorm on campus. Focused on her education, she joined the internationally well-known band, the Ocean of Soul, pledged a sorority, and studied for medical school. They shared a little bit of this time with me:

> **Winfred:** I would see her walking back and forth from her dorm, which was not far from the student center where my fraternity brothers and I were hanging out. For me, it was love at first sight.

> **Johnnie:** My sorority sister set us up on a blind date, and we went out. "Ok, well, he seems like an upstanding, nice person," so we kept dating.

> **Winfred:** I used to pick her up and take her to Aunt Batt's because they had really good stuffed shrimp. Johnny got a chance to see part of Fifth Ward.

> **Johnnie:** When I met Winfred, he had a car and he would take me to Fifth Ward. It was, obviously, an underserved area. You could see the housing just driving down the street—little small cottage-like homes, but they were mostly well-kept lawns. They didn't have sidewalks. I met his family and his extended family. I had never seen such a huge family. Sixty plus people. I really enjoyed being part of it—getting to know all the different family members. After I graduated, I went to medical school in Levitt, Texas and he would write me a love letter once a week. I still have every one of them—almost 300 different love letters.

My dad's discipline and drive to secure a more stable socio-economic existence led him to find a job as a sales manager at KTRK-TV Channel 13 in Houston for 30 years. My mother became a pediatric doctor. With good-paying jobs, they decided to purchase a home in Fondren Southwest, a middle-class area made up of several subdivisions "inside the loop" of the 610 Interstate. On the weekends, we drove about 25 (traffic-less) minutes to Fifth Ward to visit my father's family and my parents often left my brother and me overnight.

In middle school, most of my African Americans friends from Missouri City were concerned about my "blackness." I was one of those African American children who "talked white" and my friends thought I needed to learn how to be "black," not realizing that despite how I sounded, my weekends were filled with more "hood" experiences than their neighborhood. One way of learning, they suggested, was memorizing a song from the Geto Boys, a popular rap group from Fifth Ward who were consistently played on the radio at the time. Prior to my introduction to rap, I was happily raised on a steady auditory diet of Curtis Mayfield and Motown, but I wanted to fit in and be as cool as the Missouri City kids, so I started listening to the local rap and R&B radio station 97.9

The Box. I approached my "blackness" with rigorous listening in my room and when I was not taking the school bus, on some rides to school with my father. The latter involved much coaxing and complaining to my dad who wanted to listen to the crackling AM news stations or old school 1970s music. However, once he realized how catchy rap was and how heavily it borrowed from his generation's music, he—eventually—acquiesced.

Geto Boys' "Mind is Playin' Tricks on Me" was the first rap song I memorized. Scarface, Bushwick Bill, and Willie D produced music that juxtaposed sweet and groovy melodies with lyrics that spoke to the harsh urban reality of the 1980s—a time marred by crack epidemics and urban violence. It was my entrée to middle school cool, surrealism, poetic license, and imagery. Southern rap ushered in a deep appreciation for hip hop that I still have today. It is not that I did not fully appreciate Public Enemy, Queen Latifah, or NWA. Listening to Geto Boys was just different. When I put on their music, I was "coming home" in a sense. The nasal Houston twang mixed with familiar images of place and words linking neighborhoods to southern experiences infused me with a sense of pride that someone was representing me/us. The lyrics exhibited a mixture of word play, humor, misogyny, gangsterism, and a sense of a place with which I was intimately entwined. The song begins:

> I sit alone in my four-cornered room staring at candles
>
> Oh that shit is on?
>
> Let me drop some shit like this here, real smooth
>
> At night I can't sleep, I toss and turn
>
> Candlesticks in the dark, visions of bodies being burned
>
> Four walls just staring at a nigga
>
> I'm paranoid, sleeping with my finger on the trigger
>
> My mother's always stressin' I ain't living right
>
> But I ain't goin' out without a fight

I was not a gangster. I was not from Fifth Ward, but I was not a tourist either. I was a recurring guest whose storyline was always tangential to the everyday reality of the protagonists, in this case, my Fifth Ward family and their neighborhood realities. Listening to rap from other regions of the United States filled me with a distinctive pleasure and emptiness; the kind of feeling that might accompany a short-lived trip. Fifth Ward was my home away from home—a sentiment that many Houstonians can understand, since our city is so big and sprawling that across town might actually feel like a whole "other country."

## 7. Cultural Memories

On Saturday nights at my grandparents' house in Fifth Ward, I stayed up late to watch Show Time at the Apollo and G.L.O.W. (Gorgeous Ladies of Wrestling), but on Sunday morning, I still eagerly got up to put on my white church socks and patent leather Mary Jane shoes while marveling at Emmola's fabulous church crowns and wigs that accompanied her transformation into a picture of glamor and sophistication. She wore full makeup, hair, jewelry, perfume, modest heels, a dress jacket, and skirt. Dressed, we climbed into Granddaddy's truck for the ride to church where he and other elder men sat at the front of the church and purveyed over the congregation.

Deacon Albert Frazier, Jr.'s role was later honored with a marker close to the side entrance of the church. His name is on the list of deacons. Louisiana historian Leon Waters, manager of the Hidden History Tours, associates historical church markers with the contributions of people in the civil rights

and social justice movements. Similarly, my grandfather's name on the marker does not simply serve to recognize his service, but also invites investigation and wonder into his life as a leader of community healing. The marker offers an opportunity to ponder how important church can be for community well-being. See Figure 12a–c.

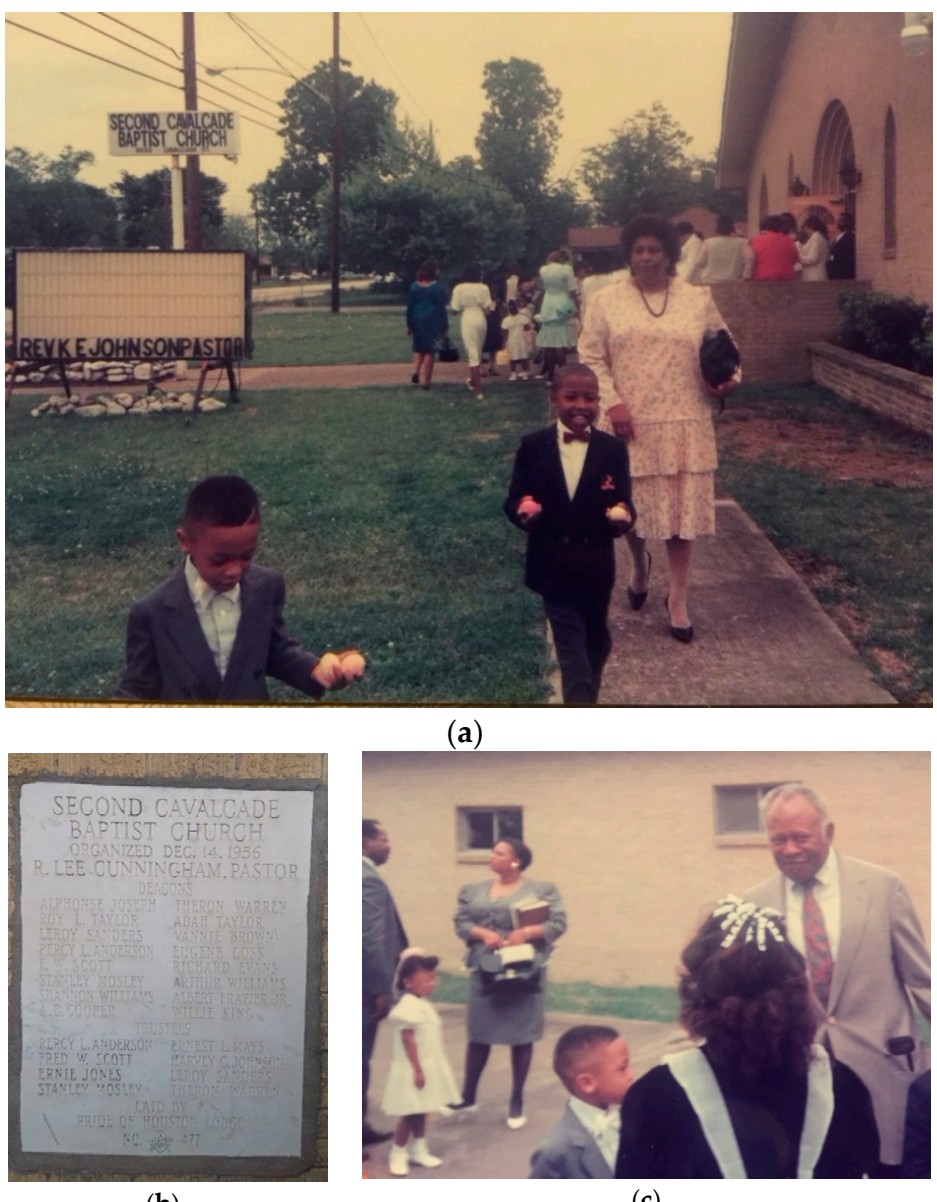

**Figure 12.** (**a**) Easter Sunday at Second Cavalcade Missionary Baptist Church. My brother and cousin with my grandmother, Emmola Frazier, not trailing too far behind; (**b**) Cornerstone of the church with my grandfather's name on it; (**c**) My grandfather, Albert Frazier, Jr., and me with my cousin, Lonnie Hill. Photographs courtesy of the Frazier family.

Back home, my grandfather had specific spaces where he sat all afternoon. When the spirit moved him to request service from someone with "younger bones," I was normally tasked with "fixin'" granddaddy's plate. Intuitively, I knew more or less what he wanted; a little bit of everything. He sat in the den and was entertained if someone sat with him but was just as peaceful if no one joined him. Meanwhile, Emmola was either cooking or resting in the kitchen. After church, or on weekends, family members and I settled in with her at the table, close to bounties of fried chicken, cakes, and pies. My mother, Johnnie, also reminds me:

I remember the tea cakes. I don't even know the origin of teacakes in African American culture, but it goes back hundreds of years ago. They are pale white cookies about the size of a flat biscuit, and they are soft inside with a semi-sweet flavor to them. It's a treat and only a few people knew how to make teacakes and make them right. It was just a matter of flour, sugar, egg, water, and maybe a drop of vanilla or cinnamon. It is a way to make a cookie with a minimal amount of ingredients, but they were really good. I still don't know today how to make them, but Sandra does.

Culinary historians trace tea cakes back to enslaved Africans in Texas and around the south who had to make do with very little. While my family does not speak of the specifics of our longer history in Texas, the cookies, as acts of love, can extend back in time (Kayal 2015). Memories of eating with my grandparents may go back even farther. More than a decade ago, while I was attending a conference in Brazil, a young Bahian woman asked me if my grandparents ate with their hands. She was making a compelling argument about West African cultural memory and how the African diaspora of the United States was, in many ways, similar to the diaspora in northeastern Brazil.

Her question still haunts me to this day. It was as if she had time-travelled and caught the quizzical look on my ten-year old face as I observed how my grandparents ate. They used utensils, but there was also a mixing of various foods and a use of the thumbs with the pointer finger and middle finger to scoop mixed flavors of cornbread with chicken and greens into a delightful bite in a way that may be similar to how West Africans use fufu. I asked my mother why my grandparents did this and was dismissed. It was as if my observation was shameful or not polite enough to prompt further inquiry. As I have gotten older, I have delighted in picking up pieces of arugula with my fingers at formal or casual dinners in public or in my own home. I frequently mix my foods, putting salad on top of beans and rice and mixing them all together. Doing so honors the cultural memory that I have retained from my grandparents and celebrates the way they enjoyed eating—with utensils, as well as with their hands.

At home on Sundays, the backyard of the house was so clearly my granddaddy's domain that I did not venture too deeply into its contents. There were tools, a shed, a truck that was consistently being worked on, a barbecue pit, and some plants in the back and occasional sunflowers. What was beyond the smoke, tools, and rusted car parts was another world—a world my granddaddy preserved from his rural upbringings in Sibley, Louisiana. He was a steel worker, but also a farmer. Granddaddy's vegetable and flower garden lined the perimeter with cherry tomatoes, squash, tall okra, sunflowers, peas, some greens, and—to my surprise—cotton. I had never imagined that someone would want to grow it in their backyard; especially my own grandfather. When he showed me the plant, I squeezed the soft white bale and was surprised by the thorny seeds in the middle. I kept thinking, "Out of all of the things he could grow, he chose cotton." Was it an almost subconscious act of repeating patterns? For my grandfather, cotton was related to a more profound cultural memory that interconnected with his agricultural upbringing and coming of age in the segregated south. His work on railroads and need to commune with the dark earth of the new urban setting allowed him to re-create himself.

In 2012, I bought a home in the Eighth Ward of New Orleans, an old Creole neighborhood close to the Mississippi River. Since Hurricane Katrina, people no longer stay long enough to be given a nickname, but a few years ago I got to know many of my neighbors when I turned a corner lot on my block from a heroin-needle, beer can, plant debris trash dump to a community garden. With the help of two friends and a few neighbors, we grew arugula, okra, tomatoes, potatoes, lettuce, sunflowers, carrots, and some herbs. For almost two years, I woke up early every morning to water the garden and care for it. We held a couple of block parties, I got to know my neighbors, and kids walking to school would wave at me or stop to look at the growing vegetables. Despite some neighbors expressing concern that we were illegally using property, the majority of people were happy with it. Sometimes I found little presents from neighbors, like stuffed animals and little notes. Every now and then, someone would mow the lawn.

In 2016, the lot was sold for $10,000 during the New Orleans adjudicated property auctions. Despite the loss of the garden, every time I plant something in dirt, I know it is partly due to the influence of my granddaddy's space in Fifth Ward. It is a reminder that home is precarious and requires attention, care, and love. Precarious, because the climate crisis and rising sea levels remind us Gulf South inhabitants that being flexible about what is "home" is a stance of survival. Home requires attention in that it is imperative to understand where someone is from in order to know who they are. Care allows us to take a place in, look around, and see who is at our sides and develop empathy for their disparate stories. And, lastly, love enables us to create homes within each other. I can look at my mother's quilt now and be happy she has it with her and think of what I will carry forward with me as well. I look forward to the day when I can once again create a beautiful garden. And I look forward to the day that I perfect my own gumbo recipe.

**Funding:** This research received no external funding.

**Conflicts of Interest:** The author declares no conflict of interest.

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
