# Peer review of "The Nickel: A History of African-Descended People in Houston’s Fifth Ward"

_genealogy, doi:10.3390/genealogy4010033_

Round 1

Reviewer 1 Report

·       Overall, this is an engaging and personable account. The article provides a complex and layered familial account of Houston, Fifth Ward, exploring generational and social mobility, economic and racial segregation and desegregation, and religion, cultural practice and community building. But a sense of clear argument is often lost. Many threads and issues are explored throughout, tied intimately to the author’s family history and sense of personal geography (for this reason the article merits high on “originality”) but these many threads could be better explicated and structured.

·       Some awkward phrasing and grammatical errors: for example: “Fifth Ward has a 43 ft. elevation, whereas other major areas Third Ward (39 ft.), Sugar Land (100 ft.) and Missouri City (79 ft.) Fifth Ward high elevation saved it from Hurricane Harvey, a slow- moving Category 4 hurricane that dumped over 27 trillion gallons of water and caused 88 deaths and thousands of people to be displaced from their homes.”

·       References for the section covering emigration to Fifth Ward (from bottom of page 2 of 20) and the discussion of the city’s historic growth (from page 3) also requires references to secondary or primary source material.

·       Phrases, rather than complete sentences (or poor choice of verbs), make some parts confusing. Half-finished sentences appear throughout. For example: “Desegregation, early white flight and African-American middle class flight to  other expanding Houston wards and neighborhoods.” This isn’t a sentence. It’s a phrase. The same applies to the following: “After feeding on a steady auditory diet of Curtis Mayfield and Motown, the Geto Boys’ Fifth Ward rap music.”

·       More on role of religion/history and religion/community building? – hinted at in page 4 of 20, line 152. Then picked up again on page 8 of 20. Perhaps this is a matter of structure and flow in the article. Could the points about religion and its role in community building/health be moved around?

·       Importance of family memories of food and hospitality – not fully explicated, or tied sufficiently to the familial and community histories of Fifth Ward. Some further explication needed here to tie these sections together. (from page 7 of 20). Picked up again on page 12 of 20 – in memories of eating. Again, perhaps this is a question of structure and flow.

·       P.12 of 20: “cultural memory traversed continents in the New World and how my family has played a significant role in this memory.” Strong and evocative statement, with implications that are not consistently explored or underlined throughout the article.

Author Response

Dear Reviewer,

Thank you so much for taking the time and consideration into reading and commenting on The Nickel. I appreciate the time and care that you took into making this piece as good as it could be.

My general response to your comments is that I wholeheartedly agree with everything written. I did not clearly convey my reasons for writing the paper. I hope the new second paragraph elucidates my reason for writing about Fifth Ward. As indicated in the manuscript, I wanted to situate a personal family narrative within a bigger historic arc of over 50 years of history.

I apologize for the grammatical errors, awkward phrases and incomplete sentences. I read over it a couple of times and corrected the sections where you noted in your review and other places where the poor grammar obscured the meaning.

I tried to group together the religious part of the piece in order to connect it with the significant role my granddad had in the church and the importance of names on church markers. This point could be explored event further, and I would be very interested in researching African-American church markers in the South and the connection with those names and civil rights or movements of resistance to white patriarchy. I did not find more research on this topic in time, so it was not flushed out as much as it could be, but that could be a follow-up piece.

The section on food and eating was not really meant to be an observance of people from Fifth Ward, but a commentary on cultural memory and an observation of the unique (or not unique) way my grandparents ate. I would love to have the ability to research eating habits, food ways, and dining culture of various families in the neighborhood, and would like to consider that for a follow-up project.

The “cultural memory” comment did not have implications that were underlined throughout the article, because the article was not about African diasporic cultural memory traversing continents to arrive in the New World. This would be a compelling argument and piece, but I hope the article conveys an observance of moments that cause wonder and curiosity about “African retentions”, and since I was only briefly analyzing the socio-economic and migratory patterns of African-Americans in the neighborhood from 1950s-2000s, I could not fully explore the idea of cultural memory and African retentions. Although, the idea is really compelling and an interest of mine.

I hope this letter addresses the comments in your review. I am very grateful to have this point of reflection in the editing process and would welcome any further dialogue on what was addressed or not addressed.

Thanks, again, for your time.

Sincerely,

Round 2

Reviewer 1 Report

Thank-you for your response - the additional paragraphs at the beginning were a nice and necessary introduction to your article, a good way to ease the reader into the "personal family narratives" that you situate alongside "unique particularities of every-day African-American life from 1960s Fifth Ward to today." 

I still think there is a bit of work to be done around sentence structure and expression, but that can be an ongoing process as you move through to the next stage of editing. There's also still that lingering question of whether the reader gets a sense of context: of changing life from "1960s Fifth Ward to today", which was one of your aims to provide. And perhaps this is also a matter of integrating some more references to the secondary historiographical literature? Overall, these are evocatively told family stories, which I hope to see published. 

Author Response

Thank you for another close reading of this piece. You are right. There were many grammatical errors that needed to be addressed. I greatly appreciate your guidance and push in making my sentence structure more clear.

In terms of the a more close investigation of 1950s, 1960s Fifth Ward, I acknowledge that this was a promise that the piece does not consider. I chose to delete it, since my research does not reflect a closer investigation into these two decades. I think this deletion does not dilute the significance of the piece. 

I can't thank you all enough for your careful consideration, and I hope this new edit brings me closer to clarity with this topic.
